# Sevoflurane Exposure of Clinical Doses in Pregnant Rats Induces Vcan Changes without Significant Neural Apoptosis in the Offspring

**DOI:** 10.3390/medicina59020190

**Published:** 2023-01-17

**Authors:** Yi Jin, Xiaoxue Hu, Fanhua Meng, Qing Luo, Henry Liu, Zeyong Yang

**Affiliations:** 1Department of Anesthesiology, International Peace Maternity and Child Health Hospital, Shanghai Jiao Tong University School of Medicine, Shanghai 200030, China; 2Shanghai Key Laboratory of Embryo Original Disease, Shanghai 200025, China; 3Shanghai Municipal Key Clinical Specialty, Shanghai 200025, China; 4Department of Anesthesiology, Guanghua Integrative Medicine Hospital, Shanghai University of Traditional Chinese Medicine, Shanghai 200052, China; 5Department of Anesthesiology, Huashan Hospital, Fudan University, Shanghai 200040, China; 6Department of Anesthesiology, Longhua Hospital, Shanghai University of Traditional Chinese Medicine, Shanghai 200032, China; 7Department of Anesthesiology and Critical Care, Perelman School of Medicine, The University of Pennsylvania, 3401 Spruce Street, Philadelphia, PA 19104, USA

**Keywords:** sevoflurane, lipidomics, toxicity, glycerophospholipid, sphingolipid RNA-seq, Vcan

## Abstract

*Background and Objectives:* Sevoflurane is a commonly used inhalational anaesthetic in clinics. Prolonged exposure to sevoflurane can induce significant changes in lipid metabolism and neuronal damage in the developing brain. However, the effect of exposure of pregnant rats to clinical doses of sevoflurane remains unclear. *Materials and Methods*: Twenty-eight pregnant rats were randomly and equally divided into sevoflurane exposure (S) group, control (C) and a blank group at gestational day (G) 18; Rats in S group received 2% sevoflurane with 98% oxygen for 6 h in an anesthetizing chamber, while C group received 100% oxygen at an identical flow rate for 6 h in an identical chamber. Partial least squares discriminant analysis (PLS-DA), ultra performance liquid chromatography/time-of-flight mass spectrometry(UPLC/TOF-MS) and MetaboAnalyst were used to analysis acquire metabolomics profiles, and immunohistochemical changes of neuronalapoptosis in hippocampus and cortex of neonatal rats were also analyzed. *Results:* This study aimed to explore lipidomics and transcriptomics changes related to 2% sevoflurane exposure for 6 h in the developing brains of newborn offspring rats. Ultra-performance liquid chromatography/time-of-flight mass spectrometry (UPLC/TOF–MS) and RNA sequencing (RNA-seq) analyses were used to acquire metabolomics and transcriptomics profiles. We used RNA-seq to analyse the expression of the coding and non-coding transcripts in neural cells of the cerebral cortex. No significant differences in arterial oxygen tension (PaO_2_), arterial carbon dioxide tension (PaCO_2_), or arterial blood gas were found between the groups. The relative standard deviation (RSD) of retention times was <1.53%, and the RSDs of peak areas ranged from 2.13% to 8.51%. Base peak chromatogram (BPC) profiles showed no differences between the groups. We evaluated the partial least square-discriminant analysis (PLS-DA) model. In negative ion mode, R2X was over 70%, R2Y was over 93%, and Q2 (cum) was over 80%. Cell apoptosis was not remarkably enhanced by TUNEL and haematoxylin and eosin (HE) staining in the sevoflurane-exposed group compared to the control group (*p* > 0.05). Glycerophospholipid (GP) and sphingolipid metabolism disturbances might adversely influence neurodevelopment in offspring. The expression of mRNAs (Vcan gene, related to neuronal development, function and repair) of the sevoflurane group was significantly increased in the differential genes by qRT-PCR verification. *Conclusions*: GP and sphingolipid metabolism homeostasis may be potential therapeutic approaches against inhalational anaesthetic-induced neurodegenerative disorders. Meanwhile, sevoflurane-induced Vcan changes indicated some lipidomic and transcriptomic changes, even if neural cell apoptosis was not significantly changed in the usual clinical dose of sevoflurane exposure.

## 1. Introduction

Each year, over 10 million pregnant women in China implement anaesthesia. Maternal sevoflurane exposure may present a possible risk for neurodevelopment in her offspring [1]. Furthermore, preclinical studies have indicated that prolonged exposure to inhalational anaesthetics during brain synaptogenesis can hurt the immature central nervous system, leading to standing neurocognitive impairments [1]. Prolonged sevoflurane inhalation decreased phosphatidylethanolamine, phosphatidylserine and phosphatidylglycerol and increased 4-hydroxynonenal (relevant to stress and lipids). Phosphatidylserine is involved in signalling pathways relevant to neural cell synaptogenesis, survival and neurite growth [2,3,4,5,6]. Alpha-lipoic acid suppresses sevoflurane-associated neural cell apoptosis by PI3K/Akt pathway [7].

In addition, sevoflurane, with high concentrations and long-term exposure, has been shown to alter the metabolism of glucose, amino acids and lipids, as well as intracellular antioxidant and osmotic substances in developmental nerves [8]. Lipid changes may disturb phospholipid homeostasis and change the integrity, orientation, permeability and function of the membrane, leading to neurological dysfunction and degeneration. Meanwhile, RNA-seq can detect alterations in the expression of low-abundance transcripts. Can sevoflurane exposure in pregnant rats induce potential neurotoxicity in offspring, including lipidomics and RNA-seq changes?

We hypothesised that sevoflurane exposure could induce lipidomic and transcriptomic changes in neurotoxicity in the developing brains of offspring rats after birth. To the best of our knowledge, this is the first study to explore lipid and transcriptomic changes in anaesthetic-induced neurotoxicity in rats. 

## 2. Methods

### 2.1. Animals 

All animal studies using rat euthanasia procedures were approved and complied with the AALAC and IACUC guidelines and guidelines for institutional animal care at Shanghai Jiao Tong University (permit number: A2016077). Sixteen Sprague Dawley (SD) rats weighing 400–500 g within 18 days of gestation were purchased from IACUC at Shanghai Jiao Tong University.

None of the rats carried viruses, bacteria or parasites. A sentinel programme continuously recorded animal health. The rats were kept in individual ventilated plastic cages (420 mm × 250 mm × 230 mm). A standard cardbox house was provided; autoclaved hay (about 8~12 g/cage) and two Nestlets™ (about 5 × 5 cm) were supplied. After high-temperature sterilisation, the rats were fed a pelleted and ordinary pure water rat diet by Jiangsu Cooperative Medical Bio-engineering Co., Ltd(Taizhou, China). The rats were housed in an environment with a temperature of 23 ± 2 °C, relative humidity of 50 ± 15%, and a light/dark cycle of 12/12 h (approximately 20 lx in the cage); air pressure was controlled at 10 Pa. High-performance liquid chromatography (HPLC)-grade acetonitrile and formic acid were purchased from Merck (Atlanta, GA, USA); a calibration solution for time-of-flight mass spectrometry (TOF–MS) was purchased from AB SCIEX (Foster City, CA, USA). Ultra-pure water was generated using a Milli-Q pure water system (Millipore, Billerica, MA, USA).

### 2.2. Experimental Protocol

SD pregnant rats, three months old, 400–500 g weighing, were randomly divided into: a sevoflurane group (S), a control group (C) and a blank group on gestational day (G) 18. Eight pregnant rats in each group were kept in a temperature-and humidity-controlled room with 12 h of light and darkness from 3:00 to 15:00. The anaesthesia chamber was given 2% sevoflurane and 100% oxygen for 6 h in the sevoflurane group. The control group was given 100% oxygen at the same flow rate in the same chamber for six hours. The blank group was stained (haematoxylin and eosin (HE) or TUNEL staining) for comparison; no oxygen or drug was performed. The parameter setting was 400 mL/min as the total gas flow. A gas analyser (Drager, Lubeck, Germany) was used to monitor oxygen, carbon dioxide and sevoflurane concentration. The rats were kept with a spontaneous breath. After sevoflurane termination, the rats were placed in a chamber containing 100% oxygen for 20 min. Pregnant mice arterial blood samples were collected and analysed using an i-STAT 1 Analyser (MN: 300-G, Abbott ABT, Chicago, USA) immediately after the termination of anaesthesia. The control group (C) received 100% oxygen at an identical flow rate for 6 h in an identical chamber. The remaining rats were kept in the chamber for delivery. The offspring of the sevoflurane group and the control group were 40 and 42, respectively. A total of 24 (7 postnatal) rats were randomly selected from offspring rats in two groups of 12 each. Serum samples of offspring rats were frozen at −80 °C for subsequent analysis.

In our study, the relevant details regarding the euthanisation of study animals were as follows. Pregnant rats were sacrificed by cutting off the abdominal aorta under sevoflurane anaesthesia or by increasing the concentration of carbon dioxide in accordance with the animal laboratory’s euthanasia method when blood was taken from the abdominal aorta directly. The newborn rats were placed in an anaesthetic box with a concentration of about 3% and then decollated under anaesthesia. Other rats were euthanised by increasing the concentration of carbon dioxide in accordance with the animal laboratory’s euthanasia method.

### 2.3. Sample Preparation and Ultra-Performance Liquid Chromatography (UPLC) Analysis

One hundred microlitres of neonatal rat serum (from sevoflurane and the control group) was placed in a tube (pretreated with heparinisation). Paclitaxel storage solution (100 µg/mL; 15 µL) was added as an internal reference material, followed by 300 μL methanol, vortex oscillation of 3 min, and high-speed centrifugation (12,000 r/min) at 4 °C for 10 min. The supernatant of precision absorption was 200 μL, and a microporous filtration membrane (0.22 µm) was used.

Chromatographic conditions: Waters Xterra MS C 8 column (2.1 × l00 mm, 3.5 μm). Mobile phase A included acetonitrile, isopropanol (5:2, *v*/*v*) containing 2 mmol/L ammonium acetate, 0.1% formic acid, and 0.1% formic acid aqueous solution containing 2 mmol/L ammonium acetate. The gradient elution procedure was as follows: 0–1 min 10% A. 1/2 min 10–30% A. 2/4 min 30/50% A. 4/8 min 50/70% A. 8/12 min 70/100%. 1/2–24 min 100% A. 24–24.5 min 100–10% A. 24.5–30 min 10% A. The flow rate was 0.35 mL/min, the column temperature was 40 ℃, and the sample volume was 10 µL.

This method was used to analyse other lipid compounds (glyceryl ester, glycerol phospholipid and high abundance sphingolipid). The related substances were identified using ultra-high-performance liquid chromatography (HPLC) with a multi-reaction monitor.

### 2.4. Pattern Recognition Analysis Based on PLS-DA

The score map was obtained using pattern recognition. The model was then evaluated, and candidate heteron ions were added to the peptide segment. The variable importance in the projection (VIP) marker in the partial least squares discriminant analysis (PLS-DA) model was used to screen for the candidate differential protein. PLS-DA was used to excavate the changes among the different samples to determine the key ingredients. After managing the data, a data matrix containing sample variables was obtained, and the next metabolite was further analysed. Compounds with projected values *p* < 0.05 and >1.0 could be identified as potential biomarkers from the PLS-DA model.

### 2.5. Hematoxylin and Eosin (HE) Staining

Sprague Dawley (SD) pregnant rats were randomly divided into the sevoflurane group (S), control group (C) and blank group on gestational day (G) 18. HE staining was used to detect neural cell apoptosis in neonatal rats’ hippocampus and cerebral cortex. For HE staining, sections were separated and then treated by hydration, haematoxylin staining, 1% ethanol hydrochloric acid alcohol differentiation for 1 s, and eosin staining. The stained slices were dehydrated, encased in neutral balm, and covered with a coverslip. The microscope (Olympus) modified the image to 400× magnification.

### 2.6. TUNEL Staining

The grouping was the same as above. Sections of the hippocampus and cerebral cortex of each group were frozen, soaked in 3% H_2_O_2_ phosphate buffer (PBS) to eliminate the endogenous peroxidase reaction, washed 3 times and 5 min once, and 20% foetal bovine serum (FBS) and 3% FBS protein were then added for 15 min. After the TUNEL reaction solution was connected with fluorescein, the slices were placed in a humidifying box at 37 °C for 1 h and washed with PBS three times for 5 min. The parts were then re-evaluated. The termination solution at room temperature acted for 10 min, and the anti-digoxin peroxidase antibody acted for 30 min at 37 °C. It was washed three times with PBS for 5 min. Afterwards, slices were made using 3,3′-diaminobenzidine, dyed using HE, made transparent using xylene, and sealed using neutral resin. The sections were then observd with a fluorescence microscope. TUNEL-positive staining was observed under an S BX 51 Upright fluorescence microscope (400-fold, USA). The experiment was repeated four times, and the apoptotic rate was calculated. The formula was as follows: apoptotic rate = TUNEL positive cells/the total cells.

### 2.7. RNA-Seq and Real-Time PCR (RT-PCR) Verification

TRIzol extracted the RNA from the neonatal rats’ cortical cells (102 to 104), including the sevoflurane group and the control group, and we added 800 µL of TRIzol reagent to the sample. After the sample was lysed, the samples were separated by adding chloroform. Before adding isopropanol to precipitate RNA, 5–10 µg RNA-free glycogen was added as an aqueous phase carrier. To reduce the viscosity of the solution, the sample was cut through the 26 needles twice before chloroform was added to cut the genomic DNA. After two-phase separation, glycogen was left in the aqueous phase and co-precipitated with RNA. The quantitative real-time (RT-PCR) reaction system consisted of 1 µL cDNA, 1 µL upstream and downstream primer mixture, 5 µL premixed solution and 3 µL actinase water. 

Follow-up database construction and sequence analysis of the full transcript were set by Cloudseq Biotech (Shanghai, China) Co., Ltd. Relevant tests for quality and RNA-seq sequencing were finished by the same company. The target gene RT-PCR was detected in the SYBR Green (TaKaRa, Basel, Switzerland) system (LightCycler 480) when needed. The instruments included the primer design software Primer 5.0 and the ViiA 7 Real-Time PCR System (Applied Biosystems, Foster City, CA, USA). The expression of related genes was administered using quantitative RT-PCR to verify significant genes, which was performed using the relevant instructions. To verify the accuracy based on the analysis of RNA-seq, the differential genes in the relevant signalling pathways were tested in the samples. Relative gene expression was calculated, and GAPDH was used as an internal reference for relative quantitative analysis.

### 2.8. Statistical Analysis

Data analyses were conducted using Graph Prism 5.0 software. Assumptions of normality and homogeneity of variance were first checked. Relevant data were expressed as mean ± standard deviation. An independent samples *t*-test was used to analyse the differences between groups for continuous measures, including TUNEL or HE staining data. Multigroup comparisons of the means were carried out with Tukey’s post hoc tests. PLS-DA was conducted in MatLab (version 3.7.1). There were statistically significant differences at a standard of *p* < 0.05. The target gene candidates of DE miRNAs were used for the Kyoto Encyclopaedia of Genes and Genomes (KEGG) enrichment analysis and pathway database [9,10,11]. Kobas [12] software was used to evaluate the statistical enrichment of the candidate genes of the KEGG pathway. We used QValue [13] to adjust the *p*-value. The Q value < 0.01 and log2 (Fold change) > 1 are set as the standard of remarkably DE. Pathways were identified to be significantly enriched when the false discovery rate (FDR) was less than 0.05.

## 3. Results

In the arterial blood gas (ABG) test, rats treated with sevoflurane in the S group and the C group were tested to eliminate the possibility of blood gas alterations. As shown in Table 1, the PH value, partial pressure of oxygen (PaO_2_), blood oxygen saturation (SpO_2_) and partial pressure of carbon dioxide (PaCO_2_) showed no significant variance between the two groups. Rats in the S group were only affected by sevoflurane but not by alterations in ABG caused by sevoflurane anaesthesia.

### 3.1. UPLC/TOF-MS Analysis of Sevoflurane-Induced Potential Neurotoxicity to Acquire Metabolomic Profiles 

UPLC/TOF–MS analysis was performed in both positive and negative ionisation modes to acquire metabolomic profiles. A representative base peak chromatogram (BPC) is shown in Figure 1A,B. Method validation was conducted on six QC samples. Six common peaks in both negative and positive ion mode (paired retention time M/Z) variances of the retention time and peach area of 10 biomarkers (Table 2) were randomly selected and further examined for validation. Furthermore, the RSDs of retention times were <1.53%, and the RSDs of peak areas ranged from 2.13% to 8.51%, indicating that the proposed method of UPLC/TOF–MS profiling was acceptable. VIP in PLS-DA is commonly used to discover relevant biomarkers of neurotoxicity. This study identified 10 endogenous metabolites as potential biomarkers (Table 2). Most biomarkers have been reported to be associated with some neurodegenerative diseases or neurodevelopment impairment (Table 2). Interestingly, these biomarkers were detected in the control (C) group but not in the sevoflurane exposure (S) group.

### 3.2. Partial Least Squares-Discriminate Analysis (PLS-DA) and Lipidic Metabolic Pathway Analysis of Sevoflurane Treatment

Principal component analysis (PCA) was performed using UPLC/TOF–MS sample profiles in both groups; nevertheless, poor separation was obtained (data not shown). Consequently, partial least squares-discriminate analysis (PLS-DA), a supervised method, was performed. A much better separation was acquired in the positive ion mode (Figure 2A). Three parameters were used to evaluate the performance of the PLS-DA model: in positive ion mode: R^2^X = 0.665, R^2^Y = 0.996, and Q^2^ = 0.962; therefore, the PLS-DA model was acceptable. The identified biomarkers were further analysed using MetaboAnalyst (http://www.metaboanalyst.ca, accessed on 18 June 2017). Briefly, glycerophospholipid metabolism was the most important metabolic pathway (Figure 2A). Furthermore, six metabolic pathways, including glycerophospholipid metabolism, glycosylphosphatidylinositol (GPI)-anchor biosynthesis, linoleic acid metabolism, alpha-Linolenic acid metabolism, sphingolipid metabolism, and arachidonic acid metabolism, were highlighted (Figure 2B).

### 3.3. HE Staining and Tunel Staining 

Observation of TUNEL and HE staining for the morphological changes in each group (×400) was administrated, and neuronal apoptosis was detected using TUNEL and HE staining from the hippocampi and cerebral cortex tissue of each offspring rat (Figure 3A,B). Cell apoptosis was detected in the hippocampus and cerebral cortex tissues of sevoflurane-induced offspring rats. However, the number of apoptotic cells in TUNEL staining was not significantly increased in the sevoflurane group compared with the control and blank groups (*p* > 0.05) (Figure 3C,D). This indicated that cell apoptosis occurred in the sevoflurane clinical dosage, but there was no significant difference in the sevoflurane group compared to the control group and blank group.

### 3.4. Vcan Gene RT-PCR Changes and Melting and Amplification Curves

Figure 4 shows Vcan gene RT-PCR changes (Figure 4A), including melting (Figure 4B), amplification (Figure 4C) and standard curves (Figure 4D). The Vcan gene provides instructions for making a protein called versican. Versican is a type of protein known as a proteoglycan, meaning that it has several sugar molecules attached. Overexpression has implications for understanding how Vcan regulates the toxicity of the cerebral cortex, including neural development, function and repair.

### 3.5. Significantly Differentially Expressed Gene Cluster Thermogram

We investigated the differential expression gene cluster map (Figure 5A). Hierarchical clustering is widely applied to gene expression data. Genes with similar expression patterns and samples with similar biological properties converge into clusters. FPKM values of significantly expressed genes obtained by comparison between groups were used for cluster analysis. mRNA scatter diagram (Figure 5B). Pearson’s correlation 0.954. Upregulated genes (640), not differential expression (10,377), downregulated genes (1810) for the sevoflurane and control groups, 640 were increased, and 1810 genes were decreased in the sevoflurane group compared with the control group; the mRNA Volcanic map (Figure 5C) showed upregulated genes (589), not differential expression (10,531), and downregulated genes (1707) for the sevoflurane and control groups. Red represents the upregulation of significantly differentially expressed genes, while green represents the downregulation of significantly differentially expressed genes. The RNA-seq experimental flow chart is shown in Figure 5D.

### 3.6. Enrichment Analysis of the KEGG Pathway

Figure 6 shows the first 20 KEGG pathways enriched by the target genes of DE miRNA in the two groups (Figure 6A), indicating that RNA transport experiences a significant change using sevoflurane exposure. The terms of gene ontology (GO) enrichment in the Up signal pathway of the DE gene exhibited significant differences between the sevoflurane and control groups. The deep signal pathway of the DE gene is shown for the sevoflurane and control groups in Figure 6B. The Up differentially expressed genes KEGG pathway (Figure 6C) included the IL-17 signalling pathway, malaria, osteoclast differentiation, pertussis, salmonella infection, spliceosome, ribosome, RNA transport, Alzheimer’s disease and a NOD-like receptor signalling pathway. The KEGG results showed that the relevant genes were expressed differently. The KEGG pathway enrichment analysis was conducted to categorise the target genes functionally. All putative target genes for DE miRNAs were enriched in the KEGG pathway. Every target gene of DE miRNAs was enriched into the KEGG pathway by functional classification of the target gene through KEGG pathway enrichment analysis. We used RNA-seq detection of transcriptomic techniques to explore gene expression to supply novel perspectives on the relevant signal pathway of sevoflurane exposure, which led to widespread and statistically significant gene changes in numerous transcripts involved in diverse pathways.

## 4. Discussion

Our data demonstrate that glycerophospholipid (GP) is a critical biomarker from GP (http://www.genome.jp/dbget-bin, accessed on 18 June 2017) in the GP metabolic process. GPs contain the main lipid category of mammalian cell membranes [14], which play an essential role in cellular functions. The deregulation of lipids is associated with Alzheimer’s, Parkinson’s and Huntington’s diseases [15]. Our previous studies demonstrated that sevoflurane induces significant damage in neural stem cells (FNSCs) [16]. Isoflurane induces cell apoptosis when used in high concentrations [17]. Sevoflurane exposure negatively influences neural development during the embryonic phase [18,19,20]. Another study showed that neurotoxicity emerges from lysosphingolipids that inhibit protein kinase C (PKC). Membrane lipids, such as gangliosides and sphingolipids, interact with soluble Aβ [21] and insoluble forms [22] and affect Aβ neural toxicity.

In our study, systems of MetaboAnalyst (http://www.metaboanalyst.ca, accessed on 18 June 2017) were used to analyse the additional pathways of the identified biomarkers. Six metabolic pathways were highlighted: glycerophospholipid metabolism, glycosylphosphatidylinositol (GPI)-anchor biosynthesis, linoleic acid metabolism, alpha-Linolenic acid metabolism, sphingolipid metabolism and arachidonic acid metabolism. Combining these methods with stable isotope labelling is suitable for analysing the metabolism of lipid species [23]. Meanwhile, we found some morphological changes in the histopathological and immunohistochemical evaluation of each offspring rat’s hippocampi and cerebral cortex tissue using HE and TUNEL staining in the sevoflurane group. However, the number of apoptotic cells was not significantly increased by TUNEL staining compared with the control group. This indicates that neural cell apoptosis is not obvious in the usual dose of clinical use, even if some lipidomic changes occur.

GPs are among the most critical lipid-forming mammalian cell membranes. Deregulation of phospholipids can lead to many diseases [24,25]. Phosphatidylcholine (PC) homeostasis is disrupted in many neurodegenerative disorders [26]. These results suggest that controlling cellular PC content, not PS content, may be useful in preventing or treating Alzheimer’s disease [27]. Disrupted phospholipid homeostasis in the endoplasmic reticulum (ER) can lead to many neurological disorders, including schizophrenia and neurodegenerative diseases such as Alzheimer’s, Parkinson’s and Huntington’s [15,28]. Sphingolipids are a diverse class of lipids composed of free sphingoid bases and their phosphates, ceramides and sphingomyelins, as well as complicated GPs [29,30], which can be detected in miscellaneous diseases, including neurological diseases [31] and metabolic disorders [32]. This study may provide a novel strategy for further exploration of the mechanism underlying inhalational anaesthetic-evoked potential neurotoxicity in the developing brain.

In our study, RNA-seq did not rely on a predesigned probe and enabled rapid profiling and deep investigation of the transcriptome for any tissue or species. GO provides a systematic language to describe the attributes of genes, which are classified by cellular component, molecular function and biological process. This study is the first report to evaluate RNA-seq changes in pregnant women for anaesthetic exposure-associated biomarkers. The qRT-PCR verification results indicated that Vcan was increased significantly in the sevoflurane group, indicating that some transcriptomic changes occurred even if neural cell apoptosis was not significantly changed in the sevoflurane group compared to the control group in clinical use. Vcan is related to nervous system development [33]. Vcan may be capable of enhancing axonal or plasma membrane viability in cultures. The complex microenvironment in the stroked brain determines which genes are translated and trafficked accordingly [34].

Four versions (isoforms) of the versican protein were produced from the Vcan gene. This versican protein likely helps regulate cell growth and division, the attachment of cells to one another (cell adhesion), and cell movement (migration). Studies suggest that versican plays a role in the formation of new blood vessels (angiogenesis) and inflammation. Versican also regulates the activity of several growth factors that control a diverse range of processes essential for cell growth. These isoforms (called V0, V1, V2 and V3) vary by size and location within the body. Versican interacts with many proteins and molecules to facilitate the assembly of the extracellular matrix and ensure its stability [34].

UPLC/TOF–MS-based lipidomics and RNA-seq provided comprehensive information for understanding the pathological process of potential neurotoxicity caused by prenatal exposure to sevoflurane. Although further evidence is required, abnormal GP and sphingolipid metabolism may be used as an alternative way to understand the mechanism underlying inhalational anaesthetic-induced neurotoxicity. Regulating glycerophospholipid and sphingolipid metabolism may be a potential therapeutic treatment for preventing inhalational anaesthetics from potentially impairing neurodevelopment. Meanwhile, the Vcan gene change in the sevoflurane group was significantly increased in the differential genes using qRT-PCR verification in the offspring’s cerebral cortex.

## 5. Conclusions

In conclusion, the results of this study suggest that lipid and RNA-seq changes are associated with sevoflurane exposure during pregnancy. They provided a better understanding of the risk in offspring exposed to sevoflurane, even if there were no significant cell apoptosis changes in the clinical dosage.

-**Question**: Can sevoflurane exposure in pregnant rats induce potential neurotoxicity in offspring, including lipidomics and RNA-Seq changes?

-**Findings**: Sevoflurane-induced Vcan changes, which indicate that some lipidomics and transcriptomics changes occurred even if neural cell apoptosis was not significantly changed in the usual clinical dose of sevoflurane exposure.

-**Meaning**: Clinical doses of sevoflurane do not induce neural apoptosis.

## Figures and Tables

**Figure 1 medicina-59-00190-f001:**
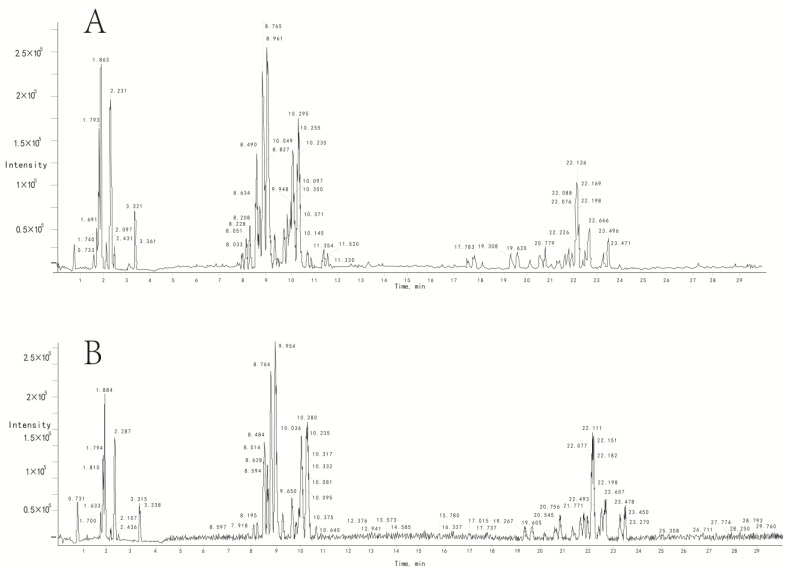
Typical base peak intensity (BPI) chromatogram of rat serum obtained in ESI positive and negative mode based on UPLC/TOF–MS analysis. (**A**,**B**) Representative BPI chromatogram of rat serum in ESI positive and negative modes, respectively. The UPLC/TOF–MS analysis was performed using an Acquity TMUPLC system (Waters Corporation) coupled to a SynaptTMG2 High-Definition Time-of-Flight Mass Spectrometry system (Waters Corporation) with electrospray ionisation (ESI) in positive and negative modes. In both the positive and negative ion modes, the BPC profiles displayed no difference between the S and C groups.

**Figure 2 medicina-59-00190-f002:**
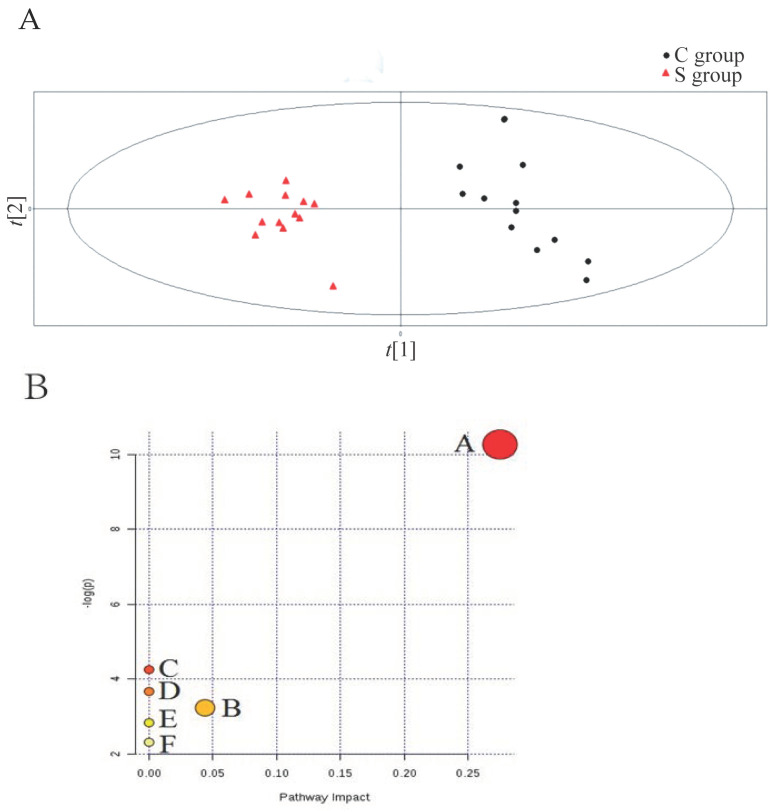
(**A**) C: PLS-DA in negative ion mode. S: PLS-DA in positive ion mode. In the C and S panels, the black boxes (.) represent samples from the C group; the open triangles (Δ) represent samples from the S group. (**B**) Pathway analysis of the biomarkers of the sevoflurane-induced neurogenerative disease model. All matched pathways were acquired according to the *p*-values from the pathway enrichment analysis and the pathway impact values from the pathway topology analysis, using the pathway library of Rattus norvegicus (rat). A. Glycerophospholipid metabolism; B. Glycosylphosphatidylinositol (GPI)-anchor biosynthesis; C. Linolseic acid metabolism; D. Alpha-Linolenic acid metabolism; E. Sphingolipid metabolism; F. Arachidonic acid metabolism.

**Figure 3 medicina-59-00190-f003:**
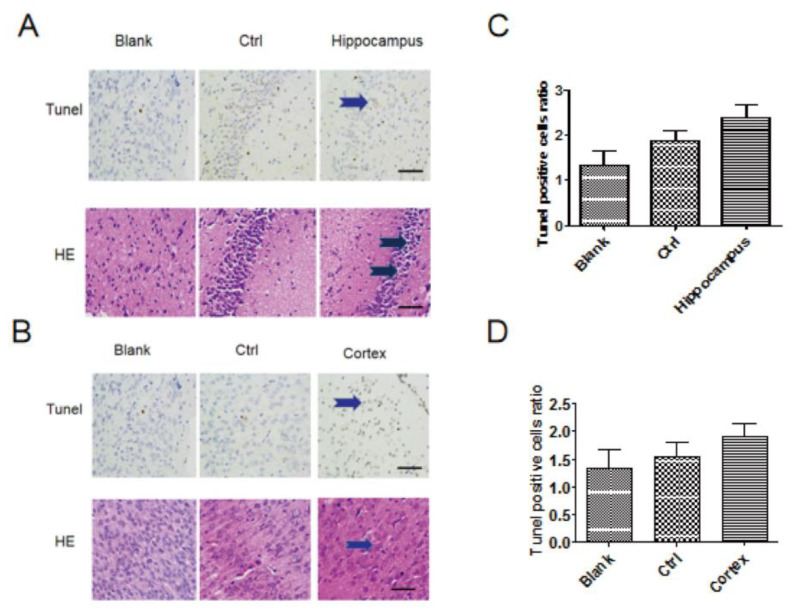
(**A**,**B**) Observation of the HE and TUNEL staining for the hippocampus and cerebral cortex tissue in morphological changes in each group (×400). (**C**,**D**) Scale bar = 200 μm. The neural cell apoptosis statistical analysis was conducted using TUNEL and HE staining from each offspring rat’s hippocampi and cerebral cortex tissue. All data were derived from the TUNEL staining results of independent experiments. Values are the mean ± SD of triplicate experiments and are analysed using one-way analysis of variance (ANOVA) with Tukey post hoc tests. There was no significant change compared to the control group.

**Figure 4 medicina-59-00190-f004:**
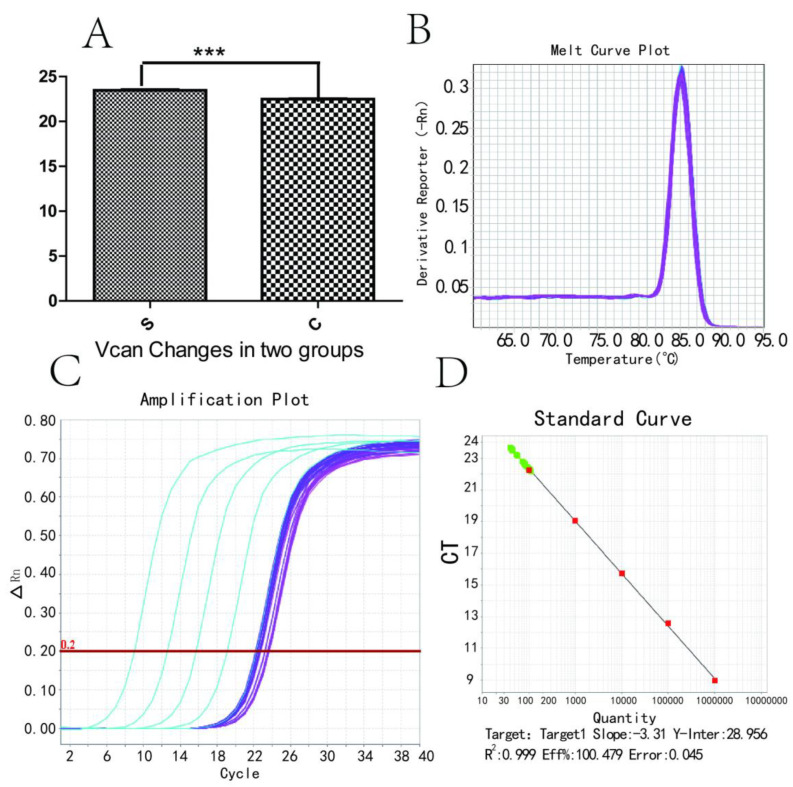
Vcan gene RT-PCR changes, including melting, amplification and standard curves. (**A**) RT-PCR changes of the Vcan gene; (**B**) melt curve plot Vcan; (**C**) amplification plot Vcan, a curve made with the number of cycles as the abscissa and the real-time fluorescence signal during the reaction as the ordinate. The amplification curve is generally divided into baseline period, exponential growth period and plateau period. (**D**) standard curve Vcan. Values are the mean ± SD of triplicate experiments and analysed using ANOVA with Tukey post hoc tests. *** *p* < 0.001 compared to the control group.

**Figure 5 medicina-59-00190-f005:**
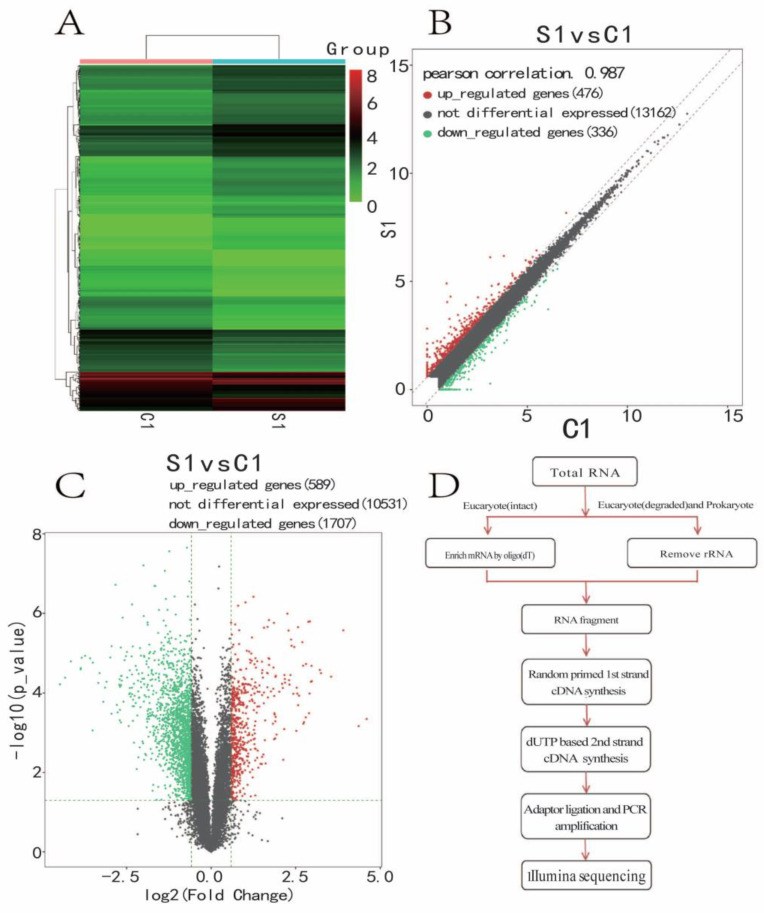
(**A**) Significantly differentially expressed gene cluster thermogram and differential expression gene cluster map. Each row represents a gene, and each column represents a sample. Red represents the upregulation of significantly differentially expressed genes, while green represents the downregulation of significantly differentially expressed genes. (**B**) mRNA scatter diagram. The *x*-axis and *y*-axis show the average FPKM value (log2 transformation) of the samples. The red dots upregulated the differentially expressed genes, the green dots downregulated the differentially expressed genes, and the grey dots represented no differentially expressed genes. Two oblique dotted lines were divided into upper, downregulated genes (1.5 times difference) and unaltered genes. (**C**) mRNA volcanic map. The x-axis represents the log2Fold_Change value, and the y-axis denotes the log10p value. The two vertical green lines were upregulated (right) and downregulated (left). The green parallel line corresponds to the *p*-value threshold. Green dots represent differentially downregulated genes, red dots represent differentially upregulated genes, and grey spots represent non-significant differentially differentiated genes. (**D**) RNA-seq experimental flow chart.

**Figure 6 medicina-59-00190-f006:**
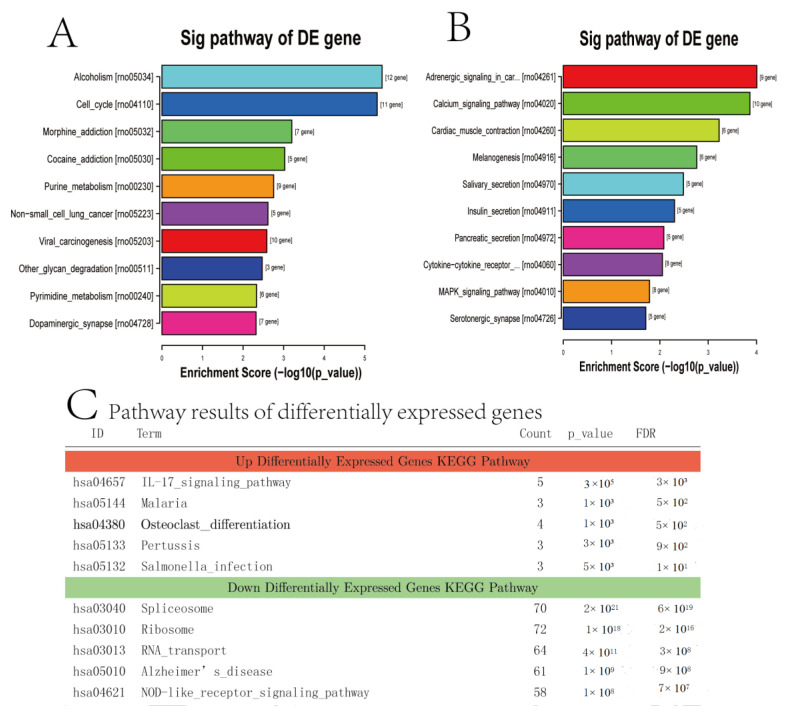
Enrichment analysis of the KEGG pathway. (**A**,**B**) The KEGG pathway analysed the top 10 item bar diagrams. In *p*-value order from low to high, the ordinate represents the *p*-value (−log10 transformation). (**C**) Pathway analysis of the first five pathways of significantly up- and downregulated differentially expressed genes KEGG pathway. Pathway analysis is a functional analysis mapping genes to KEGG pathways. The *p*-value (EASE-score, Fisher-*p*-value or Hypergeometric-*p*-value) denotes the significance of the pathway correlated to the conditions. The lower the *p*-value, the more significant the pathway (the recommended *p*-value cut-off is 0.05). Upregulated differential expression gene KEGG pathway analysis results folder: Up; downregulated differential expression gene KEGG pathway analysis results folder: Down.

**Table 1 medicina-59-00190-t001:** Arterial blood gas analysis of the two groups.

	C Group (*n* = 8)	S Group (*n* = 8)
PH	7.37 ± 0.03	7.41 ± 0.01
PaCO_2_ (mmHg)	43.80 ± 3.50	46.97 ± 4.25
PaO_2_ (mmHg)	116.0 ± 12.66	124.3 ± 19.59
BE	−0.33 ± 1.53	4 ± 1.0
HCO_3_	25.57 ± 1.75	29.90 ± 1.90
(SpO_2_)	98.33 ± 0.58	100 ± 0

Arterial blood gas analysis for the two groups. There was no significance for the two groups. PH value, partial pressure of oxygen (PaO_2_), partial pressure of carbon dioxide (PaCO_2_), HCO_3_, BE and blood oxygen saturation (SpO_2_) were monitored.

**Table 2 medicina-59-00190-t002:** Potential biomarkers and related pathways or diseases.

No.		M/Z	RT (min)	Ion	Metabolite	Related Pathways	Related Diseases
1	1687	496.341	2.27	[M+H]+	LysoPC(16:0)	glycerophospholipid metabolism **	pancreatic cancer, Barth sgl, atherosclerosis
2	2918	707.4991	2.96	[M+K]+	DG(20:1n9/0:0/20:5n3)		
3	3488	774.5638	7.86	[M+H]+	PE(22:6(4Z,7Z,10Z,13Z,16Z,19Z)/P-18:1(11Z))	glycerophospholipid metabolism **	pancreatic cancer, Barth sgl
4	3037	728.5217	8.57	[M+Na]+	SM(d18:0/16:0)	sphingolipid metabolism **	atherosclerosis
5	3929	808.5897	8.85	[M+H]+	PC(16:0/22:5(7Z,10Z,13Z,16Z,19Z))	glycerophospholipid metabolism **	
6	3897	806.5729	9.08	[M+Na]+	Galactosylceramide (d18:1/22:0)	sphingolipid metabolism **	Hidradenitis suppurativa
7	3535	780.5505	9.13	[M+Na]+	CerP(d18:1/26:0)		
8	3238	756.5559	9.8	[M+Na]+	PC(16:0/16:0)	glycerophospholipid metabolism **	Adolescent idiopathic scoliosis
9	5586	923.7476	22.92	[M+H]+	TG(20:5(5Z,8Z,11Z,14Z,17Z)/18:2(9Z,12Z)/20:5(5Z,8Z,11Z,14Z,17Z))		
10	4776	857.7568	23.98	[M+K]+	TG(14:0/20:1(11Z)/15:0)		

RT: retention time(minute). M/Z: mass-to-charge ratio. VIP: variable importance in the projection. Student’s *t*-test was performed by Graph Prism 5.0, and *p*-values were considered statistically significant if they were, ** *p* < 0.01.

## Data Availability

All data supporting our findings are included in the manuscript. The datasets used and/or analyzed during the current study are available from the corresponding author on reasonable request.

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
