# Peer review of "Sevoflurane Exposure of Clinical Doses in Pregnant Rats Induces Vcan Changes without Significant Neural Apoptosis in the Offspring"

_medicina, 2023, doi:10.3390/medicina59020190_

Round 1

Reviewer 1 Report

the topic of the study interesting and popular. But English was difficult to understand. I could not understand the method, in Figure 3 there is a blank group but I could not find it in methods section. Again in Figure 3 does the hippocampus and cortex on the HE and TUNNEL pictures stands for sevoflurane group? 

Author Response

Comments and Suggestions for Authors

the topic of the study interesting and popular. But English was difficult to understand. I could not understand the method, in Figure 3 there is a blank group but I could not find it in methods section. Again in Figure 3 does the hippocampus and cortex on the HE and TUNNEL pictures stands for sevoflurane group? 

Dear professor, we have revised them, blank group(without treatment), Blank group was stained (Hematoxylin and Eosin (HE) or TUNEL staining) for comparison, no oxygen or drug was performed. Sprague Dawley (SD) pregnant rats were randomly divided in: sevoflurane group (S) , a control group (C) and blank group at gestational day (G) 18. The anesthesia chamber was given 2% sevoflurane and 100% oxygen for 6 hours in sevoflurane group. Control group was given 100% oxygen at the same flow rate in the same chamber for 6 hours. Blank group was stained (Hematoxylin and Eosin (HE) or TUNEL staining) for comparison, no oxygen or drug was performed.

Yes, dear professor, hippocampus and cortex on the HE and TUNNEL pictures in figure 3 stand for sevoflurane group

Reviewer 2 Report

The present manuscript written by Yi Jin et al., indicates that sevoflurane exposure in pregnant female rat can induce neurotoxicity in the offspring. The paper is easy to follow and written well. And this manuscript would be helpful for understanding correlation between anesthesia during pregnancy and neurodegenerative disorders in the offspring.

Just a few comments.

Minor points:

(1)  Tabel 1: Please complete table 1, SaO2 is missing

(2)  References: alignment in the page format for references 32-35.

Author Response

Comments and Suggestions for Authors

The present manuscript written by Yi Jin et al., indicates that sevoflurane exposure in pregnant female rat can induce neurotoxicity in the offspring. The paper is easy to follow and written well. And this manuscript would be helpful for understanding correlation between anesthesia during pregnancy and neurodegenerative disorders in the offspring.

Just a few comments.

Minor points:

  • Tabel 1: Please complete table 1, SaO2 is missing

We have revised it. “SaO2” was changed with “PaO2”.

  • References: alignment in the page format for references 32-35.

  Ok! Dear professor.